# Targeting Colorectal Cancer: Unravelling the Transcriptomic Impact of Cisplatin and High-THC Cannabis Extract

**DOI:** 10.3390/ijms25084439

**Published:** 2024-04-18

**Authors:** Viktoriia Cherkasova, Yaroslav Ilnytskyy, Olga Kovalchuk, Igor Kovalchuk

**Affiliations:** Department of Biological Sciences, University of Lethbridge, Lethbridge, AB T1K 3M4, Canada; viktoriia.cherkasova@uleth.ca (V.C.); slava.ilyntskyy@uleth.ca (Y.I.)

**Keywords:** cisplatin, cannabis extract, delta-9-tetrahydrocannabinol, colorectal cancer

## Abstract

Cisplatin and other platinum-derived chemotherapy drugs have been used for the treatment of cancer for a long time and are often combined with other medications. Unfortunately, tumours often develop resistance to cisplatin, forcing scientists to look for alternatives or synergistic combinations with other drugs. In this work, we attempted to find a potential synergistic effect between cisplatin and cannabinoid delta-9-THC, as well as the high-THC *Cannabis sativa* extract, for the treatment of HT-29, HCT-116, and LS-174T colorectal cancer cell lines. However, we found that combinations of the high-THC cannabis extract with cisplatin worked antagonistically on the tested colorectal cancer cell lines. To elucidate the mechanisms of drug interactions and the distinct impacts of individual treatments, we conducted a comprehensive transcriptomic analysis of affected pathways within the colorectal cancer cell line HT-29. Our primary objective was to gain a deeper understanding of the underlying molecular mechanisms associated with each treatment modality and their potential interactions. Our findings revealed an antagonistic interaction between cisplatin and high-THC cannabis extract, which could be linked to alterations in gene transcription associated with cell death (*BCL2*, *BAD*, *caspase 10*), DNA repair pathways (*Rad52*), and cancer pathways related to drug resistance.

## 1. Introduction

Colorectal cancer (CRC) is the third most prevalent cancer globally. It is frequently diagnosed at advanced stages, thereby constraining treatment options [1]. Even with various prevention efforts and treatments available, CRC remains deadly. There is a need for new and better ways to prevent and treat it, possibly by combining different drugs. Recent research suggests that cannabinoids could be promising in this regard [2,3,4,5,6,7,8,9,10].

In recent years, both our experimental data and data from others have demonstrated the anticancer effects of cannabinoids on CRC [11,12,13,14,15,16]. Potential mechanisms through which cannabinoids affect cancer involve the activation of apoptosis, endoplasmic reticulum (ER) stress response, reduced expression of apoptosis inhibitor survivin, and inhibition of several signalling pathways, including RAS/MAPK and PI3K/AKT [2,6,11,17]. Our research has revealed that *Cannabis sativa* (*C. sativa*) plant-derived cannabinoid cannabidiol (CBD) influences the carbohydrate metabolism of CRC cells, and when combined with intermittent serum starvation, it demonstrates a strong synergistic effect [16].

In 2007, Greenhough et al. reported that delta-9-tetrahydrocannabinol (THC) treatment in vitro induces apoptosis in adenoma cell lines. The apoptosis was facilitated by the dephosphorylation and activation of proapoptotic BAD protein, likely triggered by the inhibition of several cancer survival pathways, including RAS/MAPK, ERK1/2, and PI3K/AKT, through cannabinoid 1 (CB1) receptor activation [11]. In contrast, exposure of glioblastoma and lung carcinoma cell line to THC promoted cancer cell growth [18].

Research examining the combination of CBD with the platinum drug oxaliplatin demonstrated that incorporating CBD into the treatment plan can surmount oxaliplatin resistance. This leads to the generation of free radicals by dysfunctional mitochondria in resistant cells and, eventually, cell death [19]. Recent study has demonstrated that the generation of free radicals might be enhanced by supramolecular nanoparticles that release platinum salts in cancer cells, which potentiates the effects of treatment [20]. Several other studies showed that THC, CBD, and cannabinol (CBN) can increase the sensitivity of CRCs to chemotherapy by the downregulation of ATP-binding cassette family transporters, P-glycoprotein, and the breast cancer resistance protein (BCRP) [21], resulting in the potential chemosensitizing effect of cannabinoids [22,23,24]. These data were one of the reasons why we decided to combine a DNA-crosslinking agent cisplatin, with a selected cannabinoid extract.

Cannabis extracts contain many active ingredients in addition to cannabinoids, including terpenes and flavonoids, which possibly have a modulating, so-called entourage effect on cancer cells [25]. Research conducted on DLD-1 and HCT-116 CRC lines demonstrated a notable reduction in proliferation following exposure to high-CBD extracts derived from *C. sativa* plants. Furthermore, the same extract has been shown to diminish polyp formation in an azoxymethane animal model and reduce neoplastic growth in xenograft tumour models [25]. The synergistic interaction between different fractions of *C. sativa* extract in G0/G1 cell cycle arrest and apoptosis was also demonstrated in CRC cells [26]. In contrast, full-spectrum CBD extracts were not more effective at reducing cell viability in colorectal cancer, melanoma, and glioblastoma cell lines compared to CBD alone. Purified CBD exhibited lower IC50 concentrations than CBD alone [27]. Thus, it appears that the extract composition and concentration of other active ingredients could be the modulating factors of the anti-cancer effect of cannabinoids [28].

The cannabis plant contains a variety of terpenes and flavonoids, which are biologically active compounds that may also hold potential for cancer treatment [29,30]. There are 200 terpenes found in *C. sativa* plants [31]. Here, we will review terpenes that were relevant to our study.

Myrcene, a terpene present in cannabis plant, demonstrated carcinogenic properties, leading to kidney and liver cancer in animal models [32] and in human cells [33]. However, it also demonstrated cytotoxic effects on various cancer cell lines [31,34].

Another terpene that appears in cannabis is pinene. Pinene, another terpene found in cannabis, has demonstrated the ability to decrease cell viability, trigger apoptosis, and prompt cell cycle arrest in various cancer cell lines [35,36,37,38,39,40,41]. Moreover, it can act synergistically with paclitaxel in tested lung cancer models [39]. In vivo animal models showed a decreased number of tumours and their growth under pinene treatment [42]. These data could also support the notion that whole-flower cannabis extracts rich in terpenes and perhaps other active ingredients are more potent against cancer than purified cannabinoids [43].

Cisplatin has a limited therapeutic window and causes numerous adverse effects, and cancer cells are often developing resistance to it [44,45]. To avoid the development of drug resistance, cisplatin is often employed in combination with other chemotherapy agents [46]. The formation of DNA crosslinks triggers the activation of cell cycle checkpoints. Cisplatin creates DNA crosslinks, activating cell cycle checkpoints, causing temporary arrest in the S phase and more pronounced G2/M arrest. Additionally, cisplatin activates ATM and ATR, leading to the phosphorylation of the p53 protein. ATR activation induced by cisplatin results in the upregulation of CHK1 and CHK2, as well as various components of MAPK pathway, affecting the proliferation, differentiation, and survival of cancer cells [47], as well as apoptosis [48].

Based on the extensive literature review, there is compelling evidence to warrant investigation into the efficacy of *C. sativa* extracts containing various terpenoid profiles. This exploration aims to determine whether specific combinations of cannabinoids with terpenoids could yield superior benefits in treating CRC cell lines compared to cannabinoids alone. Therefore, evaluating selected cannabinoid extracts alongside conventional chemotherapy drugs, such as cisplatin, holds promise. This approach is particularly advantageous given the prevalence of cancer patients using cannabis extracts for alleviating cancer-related symptoms. Here, we analyzed steady-state mRNA levels in the HT-29 CRC cell line exposed to cisplatin, high-THC cannabinoid extract, or a combination of both treatments.

## 2. Materials and Methods

### 2.1. Chemicals Used

Cisplatin was purchased from Sigma-Aldrich, Darmstad, Germany (CAS 15663-27-1). THC was obtained from Sigma-Aldrich (Cerilliant, T-108). Dimethyl sulfoxide anhydrous (DMSO) was obtained from Thermo Fisher Scientific, Waltham, MA, USA (Cat#D12345). Cisplatin stock solution (100 μM) was dissolved in DMSO and kept at −20 °C for two weeks maximum.

Activated *Cannabis sativa* extracts, including extract #18, were prepared from dried flowers. *Cannabis sativa* plants were provided by the “Pathway Rx” and “Sundial Growers” companies. Stocks were prepared in a 60 mg/mL concentration and a DMSO concentration of 0.25% and kept at −20 °C.

### 2.2. Cell Lines Used in the Experiments

Three human CRC cell lines, namely, HT-29 (HTB-38™), HCT-116 (CCL-247™), and LS-174T (CL188™), were used for the experiments. The CRC cell lines HCT-116 and LS-174T were purchased from ATCC (Rockvile, MD, USA), while the HT-29 was gifted by Dr. Roy Golsteyn (University of Lethbridge).

### 2.3. Treatments

Treatment with cisplatin and THC. IC50s was calculated after exposing cells to 1–15 μM of cisplatin and THC; all treatments were conducted by diluting cisplatin or THC in the fresh complete media.

Treatment with cannabis extracts. Cannabis flowers were extracted using ethanol under the license to Dr. Igor Kovalchuk. After evaporation, the resin was dissolved in DMSO to achieve the concentration of 60 mg/mL; extracts were stored at −20 °C. To treat cells, extracts were diluted to the desired concentration.

### 2.4. Terpene Analysis

The terpenes of extract #18 were analyzed using dry flowers with the help of 8610C GC coupled with a flame ionization detector (SRI Instruments at Canvas Labs, Vancouver, BC, Canada).

Based on the HPLC data, extract #18 was high in THC (~10%) and low in CBD (<1%). As indicated by the terpenoid profile, extract #18 had high levels of β-myrcene and α-pinene (refer to Table 1).

### 2.5. High-Performance Liquid Chromatography (HPLC) and Mass Spectrometry Analyses

HPLC and mass spectrometry were performed to detect the levels of cannabinoids in cannabis extracts. For separation, the Phenomenex Kinetex EVO C18 column with a Phenomenex SecurityGuard ULTRA guard column were used. Standards calibration and sample analysis were carried out using 2 μL of the injection volume. Compound peaks were detected at 230 nm and 280 nm wavelengths. Mobile phases included 50 mM ammonium formate (Sigma-Aldrich) in HPLC grade water on the A side and 100% methanol on the B side. The flow rate was 0.3 mL/min.

Analysis was conducted on two independent samples, with two technical repeats per each sample. The crude extracts were resuspended in 1 mL of methanol and filtered through a 0.22 mL syringe PTFE filter. The filtered extract was diluted ten times with methanol and then separated by HPLC. The extract was fractionated in the fraction collector (Agilent Technologies serial #DE63056961), and nine fractions were collected according to the obtained chromatogram. CBD and THC purchased from Sigma were used as external calibration standards.

### 2.6. Cell Viability Assay

MTT [3-(4,5-dimethylthiazol-2-Yl)-2,5-diphenyltetrazolium bromide] assay was used to measure cell viability in response to cisplatin, THC, and cannabis. Cells were grown to !80% confluency and then trypsinized by TRYPSIN/EDTA (0.25% Trypsin and 2.21 mM EDTA-4Na; Cat#325-043-EL; WISENT Inc., Saint-Jean-Baptiste, QC, Canada). After trypsinization and centrifugation, fresh media were added to the cells, and one 1000 cells per well were plated into flat-bottom 96-well plates in 100 μL of the appropriate medium; three replicates were carried out for each experimental unit; cells were incubated overnight before any treatment was performed. Cells were treated for five days, with daily changes in cisplatin, extract, THC, or DMSO. Incubation temperature was 37 °C in the presence of 5% CO_2_ for 24 h.

To evaluate cell viability at every experimental endpoint (24, 48, 72, 96, and 120 h of treatment), 10 μL of MTT kit I (#11465007001, Roche, Mississauga, ON, Canada) was added, and these plates were further incubated at 37 °C in the CO_2_ incubator for four hours. After the incubation, 100 μL of MTT solution was added to each well, and the plate continued to incubate at 37 °C overnight. To measure cell viability, the absorbance was measured at 595 nm using a FLUOstar Omega microplate reader (BMG Labtech, Ortenberg, Germany). The results were calculated by relating the treatments to the appropriate controls. All treatments were conducted in triplicate, and each test was performed thrice.

### 2.7. RNA Extraction and Gene Expression Analysis of the HT-29 Cell Line

HT-29 cells were grown to confluency of ~80%, then detached using TRYPSIN/EDTA (0.25% Trypsin and 2.21 mM EDTA-4Na, Cat#325-043-EL, WISENT Inc., Quebec, Canada) and re-plated at a density of 6 × 10^5^ cells per well. After 24 h, cells were treated for 72 h, refreshing treatments every 24 h. Cells were harvested by washing them twice with ice-cold PBS and the TRIzol™ Reagent (Invitrogen, Carlsbad, CA, USA). Total RNA was extracted with the TRIzol Reagent as per manufacturer protocol. The quality and quantity of the extracted RNA were analyzed using a NanoDrop 2000/2000c (Thermo-Fisher Scientific Company, Wilmington, DE, USA).

The RNA samples were shipped to Génome Québec (Montréal, QC, Canada) for library preparation and sequencing. The stranded mRNA libraries were prepared using NEBNext^®^ Ultra™ RNA Library Prep Kit for Illumina^®^ (New England BioLabs, Ipswitch, MA, USA). Sequencing was performed using Illumina NovaSeq6000 (San Diego, CA, USA) S4 PE 100 bp—25 M reads.

### 2.8. Pathway Analysis and Bioinformatics Workflow for the HT-29 CRC Cell Line

Bioinformatics workflow. Human GRCh37 (Ensembl) downloaded from Illumina iGENOME was used as a reference. Base calling and demultiplexing were conducted using Illumina CASAVA 1.9 pipeline.

Initial sequencing library quality control was analyzed with FastQC 0.11.8. Reads were mapped to the human genome (Ensembl, GRCh37) using hisat2 version 2.0.5. Read mapping to genes was counted using featureCounts version 1.6.1 from the Subread package. Additional summary statistics and additional quality control were performed using MultiQC [49].

The exploratory analysis included non-supervised hierarchical clustering and principal component analysis (PCA). Raw count data was loaded into R version 3.6.1. Genes with a low expression level defined as less than 1 count per million in at least 2 samples were removed from the analysis. Normalization and variance-stabilizing transformation were applied to the raw count data as described in the DESeq2 manual [50]. Relationships between samples were explored using non-supervised hierarchical clustering and PCA. The results of the exploratory analysis were visualized as heatmaps and principal component plots. Clustering and PCA were based on the top 1500 of genes with the highest median absolute deviation (MAD). In the case of hierarchical clustering, the distance measure was Euclidean and the clustering algorithm—ward.D.

We expected the biological replicates to cluster closely in the PCA plot, which reflected the similarity of their gene expression profiles. Since inter-individual variance is not expected within the same cell line, the biological replicates belonging to separate clusters in the case of cell lines indicate a high degree of technical variance. All samples clustered perfectly according to the experimental treatments. No samples were excluded as statistical outliers.

Differential expression analysis. Raw count values for each sequencing sample were loaded into R version 3.6.1. Normalization was conducted using the DESeq function with default parameters. Differentially expressed genes were detected using DESeq2 1.24.0 Bioconductor package as described in the package vignette. Bonferroni–Hochberg adjustment for multiple comparisons was performed, and genes with adjusted *p*-values < 0.05 and over 1.5-fold increase or decrease in expression were selected as differentially expressed. The results were annotated with gene symbols, entrez ids, and gene descriptions using biomaRt 2.40.3 Bioconductor package [51,52].

Over-represented gene ontology (GO) terms were detected using TopGO v2.36.0 Bioconductor package [53]. Over-represented Kyoto Encyclopedia of Genes and Genomes (KEGG) pathways were detected using GOstats v2.50.0 Bioconductor package [54]. Lists of differentially expressed genes (adjusted *p*-value < 0.05 and log2FC > 0.59) detected with DESeq2 were used to find significantly over-represented GO categories and KEGG pathways. The results were saved in each of the comparisons sub-folders. Over-represented GO categories were detected using the topGO 2.36.0 Bioconductor package. GO term analysis was conducted against the biological process (BP) category using an elimination test to focus on the most specific terms. Significantly enriched KEGG pathways were detected using GOstats 2.50.0 Bioconductor package and GAGE 2.34.0 Bioconductor packages [55]. Pathview 1.24.0 Bioconductor package [56] displayed the top 30 pathways in uni-directional GAGE analysis. The uni-directional (up- or downregulation) GAGE analysis results are saved in the files with the following naming scheme: **vs**_kegg.p.txt. Bi-directional analysis detects perturbed pathways with no regard to the overall direction of the change. The results of the bi-directional analysis are saved in **vs**_kegg.2d.txt.

R plotting functions were used to visualize the date. The results were shown as MA, volcano plots (ggplot2_0.9.3.1) and heatmaps (the heatmap.2 function from gplots package 2.13.0)–. Heatmaps showing the pathway analysis results were output using the SigGeneSet function with the option ‘heatmap=TRUE’. Bioconductor package Pathview (version 1.2.4) was used to draw significantly enriched pathways.

GeneCards—the human gene database www.genecards.org—was used for the pathway analysis [57,58].

### 2.9. Statistical Analysis

Significant difference between treatments for the analysis of cell viability experiments was calculated in GraphPad Prizm 9.0 software using one-way and two-way analysis of variance (ANOVA), as well as the two-tailed Student’s *t*-test. The *p* < 0.05 was considered statistically significant. The results were shown as the mean with standard deviation (SD).

### 2.10. Calculation of Combination Index (CI)

Drug combination analysis was performed using the Chou–Talalay method using CompuSyn 1.0 software to evaluate synergism, additivity, or antagonism [59,60,61]. In Chou’s approach, the scattering data points fit the median-effect principle with mass-action law. The combination index (CI) was calculated to quantify synergism and antagonism of the drugs; CI < 1 indicates synergism, CI = 1 indicatesadditive effect, and CI > 1 indicates antagonistic effect.

## 3. Results

### 3.1. High-THC Cannabinoid Extract Has an Inhibitory Effect on Colorectal Cell Lines

We tested more than forty whole-plant C. sativa extracts on the HT-29 CRC cell line using cell viability assay and selected extract #18 for further experiments (data are shown for extract #18 only, see Figure 1).

To establish the time and dose-dependent effects of treatments MTT assay was performed and the IC50 concentrations established for all three CRC cell lines (Figure 1). We observed a time and dose-dependent impact of cisplatin, extract #18, and THC alone on all examined cell lines during treatment. Cell viability was notably suppressed across the tested concentrations, particularly for cisplatin and extract #18.

Our results suggested that extract #18 was more effective in inhibiting CRC cell growth than THC alone (see Figure 1). Next, we explored whether combining cisplatin with extract #18 could produce a synergistic effect.

Subsequently, we conducted a combinatorial drug screen involving extract #18 and cisplatin on three CRC cell lines (Figure 2). According to the cell viability analysis, the combination of extract #18 with cisplatin exhibited antagonistic interaction, except for the higher dose treatments on the HT-29 CRC cell line. Additionally, the dose reduction index (DRI) values for cisplatin were higher than 1 in the HCT-116 and the HT-29 cell lines, indicating the possibility of cisplatin’s dose reduction, which potentially could have clinical implications.

To elucidate the underlying impacts of cisplatin on the HT-29 CRC cell line, we conducted mRNA sequencing and bioinformatics analysis to identify pathways affected by cisplatin, extract #18, and their combination (Figure 3). The unsupervised clustering based on the underlying data determined if there were sub-categories within DMSO control and the treatments. As the heatmaps showed, there were significant differences in gene expression between treatment groups. The differences were not substantial for the biological replicates within the same group, indicating the good quality of the treatment samples. For our further work, we selected the HT-29 CRC cell line, which closely mirrors the classical subtype of CRC. This cell line harbours mutations in *APC*, *PI3K*, and *P53* genes, which are frequently observed in CRCs.

### 3.2. Cisplatin Changed the Transcription of Genes Responsible for Cell Metabolism, Translation, Cell Adhesion, and Cytokine Responses in the HT-29 Cell Line

Cisplatin decreased the expression of several genes related to glycolysis, gluconeogenesis, and the citrate cycle pathways (Figure 4A,B) in the HT-29 CRC cell line compared to the DMSO control. This indicates the potential metabolic effects of cisplatin on the tested CRC cell line, which may deplete the energy resources of cancer cells and drive them toward cell cycle arrest or even cell death. Additionally, based on the mRNA sequencing results, cisplatin downregulated the expression of multiple genes involved in translation processes, including amino-acyl tRNA synthesis, ribosome synthesis, and mRNA transport, maturation, and the initiation of translation (see Figure 4C–E) compared to the DMSO control. These findings suggest that cisplatin may activate stress pathways leading to global translation inhibition, reducing energy expenditure, and potentially enhancing cell survival.

Cisplatin induced upregulation in the expression of numerous genes involved in cytokine–cytokine receptor interaction compared to the DMSO control. As depicted in Figure 5A, there was a significant increase in mRNA expression for chemokines *CCL22* from the CC and CXC subfamilies, accompanied by a decrease in *CXCR4* receptor expression. Furthermore, cisplatin upregulated the expression of genes encoding receptors for interleukins IL2, IL4, IL7, IL9, IL15, IL21, and IL13. There was also an increase in the expression of genes encoding multiple class II helical cytokine receptors, including those binding to IL10, IL22, IL26, IL28 A and B, and IL29. Additionally, genes encoding IFNAR1, IFNAR2, and IFNGR1, interferon family receptors, showed increased expression. Although increased expression of *TGF-β* gene could indicate potential cancer cell invasiveness, most TGF-β receptor genes were decreased, suggesting that cancer cells may not undergo epithelial–mesenchymal transition (EMT) and achieve invasive potential.

In the HT-29 cell line, cisplatin downregulated *Ras* and *PI3K* expression while upregulating *p21*, which may lead to cell cycle arrest, the inhibition of proliferation, and the suppression of cell survival mechanisms (Figure 5C). However, it increased JAK/STAT signalling, potentially inhibiting apoptosis and promoting cell cycle progression through the upregulation of *cyclin D* expression.

Cisplatin also increased the expression of genes encoding most lysosomal acid hydrolases, membrane proteins, and ATPeV, indicating heightened lysosomal activity in cancer cells potentially leading to cell death (Figure 5B). Furthermore, it upregulated multiple genes encoding proteins involved in phagosome formation, as well as major histocompatibility complex I (MHCI) and MHCII, responsible for antigen presentation and recognition by immune cells (Figure 5D).

In Figure 5E, representing natural killer (NK) cell-mediated cytotoxicity, cisplatin increased the expression of genes encoding MHCI, IFN receptors, and integrins present on stressed and tumour cells, potentially attracting cytotoxic immune cells. Additionally, there was an increase in the expression of proapoptotic genes such as *ERK1/2*, *TRAIL*, *Bid*, and *caspase 3*, alongside the downregulation of *Rac*, *PI3K*, *Ras*, and *PKC*, contributing to cisplatin’s antiproliferative and proapoptotic properties.

Moreover, cisplatin strongly inhibited oxidative phosphorylation by targeting all five complexes, leading to a substantial reduction in ATP production in cancer cells (Figure 5F). Furthermore, it activated the ER stress response via PERK and IRE1α and increased the expression of pro-apoptotic genes, including *Bid*, *Bad*, *Apaf1*, and *caspases 8*, *9*, *7*, and *3*.

Cisplatin exhibited an upregulation of various molecules involved in cell adhesion, potentially indicating its role in stimulating EMT and tumour cell invasion. Notably, there was an increase in the expression of genes encoding PDL-1 receptors, which are targeted in cancer immune therapies, suggesting a potential enhancement in the therapeutic effectiveness of monoclonal antibodies targeting PDL-1 (Figure 6A).

Furthermore, cisplatin upregulated genes encoding clathrin and other proteins associated with endo- and exocytosis enhancement in the HT-29 CRC cells (Figure 6B). This was accompanied by the downregulation of Ca signalling via PLC and the upregulation of pro-apoptotic proteins, except for PUMA. Additionally, cisplatin increased the expression levels of genes encoding antioxidants such as superoxide dismutase 1 and 2 while decreasing the expression of *PPARγ* gene.

Under the influence of cisplatin, there was an upregulation of *EGFR* gene, indicating a potential autocrine signalling cascade in cancer cells essential for self-initiated proliferation and activation of the MAPK cascade (Figure 6C). Conversely, there was an increased transcription of *ZO-1* encoding a protein responsible for maintaining tight junctions in cancer cells, potentially preventing cancer invasion and metastasis of the HT-29 CRC cells. Moreover, cisplatin upregulated the MAPK pathway through the increased expressions of *p38*, *MKK4*, and the transcription factor *AP-1* genes.

Figure 6D illustrates cisplatin’s impact on multiple pathways involved in the immune system response. It showed an increase in the expression of genes encoding MHC class II responsible for antigen presentation, ICAM1-enhancing cellular adhesion, and cytokines responsible for inflammatory reactions such as IL-1β and IL-8. There was also an upregulation of *TGF-β* gene, potentially indicating the activation of EMT, and VEGF, responsible for angiogenesis, which could promote the invasive and metastatic potential of the HT-29 cancer cells. TGF-β and IL-23 may stimulate Th17 cells, inhibiting the cellular immune response of cancer cells. Notably, cisplatin downregulated the toll-like receptor signalling pathway, TLR 2/4, along with IL-18, which is required for killer T-cell activation, potentially indicating an attempt by cancer cells to avoid immune destruction.

Overall, despite cisplatin’s ability to activate programmed cell death and cell cycle arrest, there was also the activation of transcripts potentially responsible for tumour progression and the development of drug resistance.

### 3.3. High-THC Cannabis Extract Altered the Transcription of Multiple DNA-Repair Systems Effectors and Cell Cycle Arrest Genes in the HT-29 Cell Line

Next, we analyzed mRNA expression pathways in HT-29 CRC line in response to extract #18.

As illustrated in Figure 7A, high-THC cannabinoid extract #18 induced multiple alterations in the expression of genes involved in tight junctions within the HT-29 cell line. The reduced expression of *claudin* gene and its downstream signalling proteins suggests a potential inhibition of cell polarity and cancer cell proliferation. Conversely, the upregulation of the *occludin* gene [62,63] may regulate paracellular diffusion and decrease paracellular permeability, although the increased expression of *c-Jun*, a component of the AP-1 complex, could promote cell survival and tumorigenesis [62,63]. Additionally, decreased levels of *PCNA* and *ERBB2* mRNA may further suppress cell proliferation and differentiation.

Similarly, the high-THC cannabinoid extract#18 downregulated genes involved in adherent junction formation in the HT-29 CRC cells (Figure 7B). This downregulation affected the expression of genes encoding proteins involved in desmosomal and adherent junctions, as well as some integrins, actin, and laminin. These changes suggest a potential loss of contact inhibition in cancer cells. However, decreased levels of integrins and laminins, which contribute to cancer cell invasiveness, may indicate the inhibition of tumour invasion. Additionally, extract #18 inhibited the expression of the gene-encoding *TCF*/*LEF* transcription factor, which is crucial in the Wnt signalling pathway. Given the HT-29′s mutation in the *APC* gene and the potential upregulation of Wnt signalling, the addition of extract #18 may alter the oncogenic signalling of this pathway, potentially exerting an anti-cancer effect.

Furthermore, the high-THC cannabinoid extract #18 decreased the expression of *PI3K*, a component of the Ras/PI3K/AKT signalling pathway often upregulated in CRCs (Figure 7C). Considering the mutation in PI3K in the HT-29 cell line, this decrease suggests a potential anticancer effect of the cannabinoid extract. Moreover, extract #18 inhibited *fibronectin* expression, likely reducing extracellular matrix interactions in cancer cells. Additionally, it inhibited various regulators of cytoskeletal dynamics and cell migration, as well as signalling pathways responsible for chemotaxis in the cancer microenvironment. These effects may imply that extract #18 can inhibit cancer cell invasion and migration, potentially preventing metastasis formation in the classical subtype of CRC.

Similar to cisplatin action, high-THC cannabinoid extract #18 significantly decreased the expression of genes encoding complexes I to V, enzymes crucial for oxidative phosphorylation (Figure 7D). Additionally, it increased the expressions of *p53*, *mPTP*, *cytochrome C*, and *caspase 9* genes, indicating the activation of apoptosis in the HT-29 CRC cell line. Furthermore, the treatment strongly decreased the expression of polymerase II, which is involved in gene transcription. The extract also downregulated genes involved in vesicular transport, endo- and exocytosis, which could prevent metastasis formation. Finally, there was a strong downregulation of *PPARγ* gene, essential for insulin sensitization and glucose uptake, potentially impairing multiple metabolic pathways utilized by cancer cells for growth and proliferation. This coincides with our previously published data, where we found that CBD changed the expression of PPAR transcripts and the altered metabolism of CRC cells [16]. Thus, the addition of extract #18 might disrupt cancer metabolism and promote self-destruction.

High-THC cannabinoid extract #18 decreased the expression of most of the genes in the short and long patch base excision repair (BER) pathway (Figure 8A). There was a downregulation of bifunctional and monofunctional glycosylases. The tested extract also decreased the expression levels of genes encoding AP-endonucleases, proteins responsible for gap filling, strand displacement, and ligation, including PCNA and polymerases δ and ε. Such a strong inhibition of BER may indicate one of the major mechanisms of anticancer effects of cannabinoid extracts in the HT-29 cell line.

There also was a significant decrease in the expression of most of the genes involved in the mismatch repair (MMR) pathway in response to extract #18 (Figure 8B). There was a lower expression of *MSH6*, *MSH2*, and *RFC* genes responsible for recognizing bulky DNA lesions. Genes encoding exonuclease I, polymerase δ, and ligase I were also downregulated, which likely disrupted the excision, DNA re-synthesis, and ligation steps of MMR.

Exposure to extract #18 resulted in changes in the expression of multiple genes responsible for homologous recombination (HR) (Figure 8C). Although we observed the up- and downregulation of genes, it is likely that the HR was overall suppressed. The expression data showed an increase in *ATM*, *Rad50*, *Rad51B*, *Rad51C*, and *BRE*, the genes needed in the initial stages of HR, including double-strand break recognition and the formation of MRX complexes and filaments. However, the expression of genes encoding downstream proteins responsible for strand invasion, DNA synthesis, strand displacement, flap removal, and ligation, such as RPA, Rad51, Rad54, polymerase δ, and BLM, was decreased.

Thus, the expression data showed that extract #18 inhibited the transcription of almost every DNA damage response pathway in the HT-29 CRC cell line. It may be one of the major mechanisms of anticancer effects of cannabinoids.

High-THC cannabinoid extract #18 significantly decreased the expression of most of the genes that participate in cell cycle regulation, with a few exceptions (Figure 9A). The decrease in the expression of several activators of cell cycle progression and increased expression of genes encoding tumour suppressors such as p53, p21, p15, and APC/C would cause cell cycle arrest at every checkpoint. First, the downregulation of *cyclin D*, *E*, and *A* together with *CDK2* genes would cause arrest in the G1/S stage, lowering the protein synthesis necessary for DNA replication. Then, the downregulation of *cyclin A*, *B*, and *CDK1* genes could stop the cell cycle in the G2 checkpoint. Lastly, the decreased expression of *cell division control 20* (*Cdc20*) and *separins* genes would lead to the arrest in the M checkpoint, being catastrophic for cancer cells.

Our data also indicated increased expressions of *c-Myc*, *E2F4* and *E2F5* genes, encoding transcription factors, which might be an attempt by cancer cells to stimulate their growth. However, the downstream effectors of those proteins were downregulated, thus not allowing to promote the growth of the HT-29 cell line, and our MTT results concur.

Cell cycle arrest gives cancer cells time to repair DNA under genotoxic stress. However, we also have seen the inhibition of DNA repair pathways, which would not let the HT-29 cells recover and might push them toward apoptosis.

High-THC cannabinoid extract #18 significantly decreased the expression of most of the genes that participate in DNA replication (Figure 9B). The expression of genes coding for DNA polymerase α-primase complex and DNA polymerases δ and ε complexes was decreased. Furthermore, the expression of genes encoding PCNA, helicase (MCM complex), RPA, and DNA ligase were also decreased, which indicates that DNA replication is strongly inhibited in the HT-29 CRC cell line. These results further support the notion that the tested extract has a major effect on cell cycle progression.

Extract #18 decreased the expression of multiple cancer signalling pathways, as presented in Figure 9C. Among downregulated genes, there were *EGF*, *EGFR*, and *ERBB2*; these genes are often overexpressed in some cancers. Also, genes encoding oncogenes EML4-ALK, Ras, ERK, PI3K, PKB/Akt, and PKC mRNA formation were downregulated. Thus, the tested extract inhibited Ras, MAPK, and PI3K cascades, the major signalling pathways regulating cell growth, survival, and apoptosis. Additionally, pro-apoptotic, *p53*, *caspase 9*, and *Forkhead* genes increased their expression. However, the decreased expressions of *E2F*, *Bax*, and *Bak*, along with the increased expression of *p21* gene, might push cancer cells toward cell cycle arrest in G1/S instead of cell death.

High-THC cannabinoid extract #18 significantly decreased the expression genes encoding PPARγ and retinoid acid receptor RXR, which regulate lipid metabolism, inflammation, cell differentiation, and apoptosis (Figure 10A). Interestingly, *PPARα* and *PPARβ/δ* expression was increased with many of their target genes that regulate ketogenesis, lipid transport, lipogenesis, cholesterol metabolism, adipocyte differentiation, and gluconeogenesis also being upregulated. On the contrary, the expression of some genes responsible for fatty acid oxidation was decreased. These data may indicate that tested cannabinoid extract increased lipid biosynthesis and decreased lipid catabolism. Thus, it increased lipid formation and storage in cancer cells via PPARα and PPARβ/δ. However, the treatment decreased the levels of transcripts of genes encoding proteins that take part in glucose metabolism in the HT-29 CRC cell line via the inhibition of PPARγ.

Extract #18 significantly increased genes encoding proteins involved in protein processing in the endoplasmic reticulum (ER), which could trigger the accumulation of misfolded proteins, ER stress, and unfolded protein response (Figure 10B). As a result, the strong upregulation of *ATF4* and *CHOP* genes could lead to pro-apoptotic signalling by downregulating *Bcl2* gene. In parallel, there was a decreased expression of *Bak*/*Bax* and *calpain* genes, which might have an anti-apoptotic effect and give the cancer cells time to recover from ER stress. This could be one of the unwanted effects of the tested cannabinoid extract on CRC cancer cells that could promote their survival.

### 3.4. The Combination of High-THC Cannabinoid Extract and Cisplatin Leads to Altered Transcription Patterns of Genes Crucial for Inducing Cancer Cell Death and Cytokine Response

Next, we analyzed the transcriptional changes in extract #18 and cisplatin combinatorial treatment in the HT-29 CRC cell line.

The combination of cisplatin with extract #18 exhibited an inhibitory effect on oxidative phosphorylation, as depicted in Figure 11A. This combination therapy resulted in decreased expression across all five complexes and hindered ATP production in the HT-29 cancer cells. Additionally, the combination treatment led to an elevation in the expression levels of genes associated with apoptosis. Notably, there was a significant increase in the expression of *ERK1/2*, *Fas/TNFR*, *Bad*, and *caspases 9*, *7*, and *3* genes. Furthermore, the upregulation of genes involved in ER stress responses, such as *PERK*, *PSEN*, and *IRE1α*, could potentiate apoptosis signalling. Interestingly, a decrease in *GAPD* expression was observed, suggesting reduced energy production in cancer cells. Moreover, there was an increase in the expression of the pro-inflammatory cytokine *IL-1* gene, which may represent an undesirable effect of the treatment, given the known role of inflammation in promoting CRC carcinogenesis.

Figure 11B illustrates the dysregulation of multiple oncogenic-stimulating pathways under the combination of cisplatin with high-THC cannabinoid extract #18 in the HT-29 CRC cell line. This combination treatment led to an increase in the expression of macrophage-colony stimulating factor (*M-CSF*), *IL-1*, and *IFNAR*, potentially promoting macrophage proliferation, differentiation, and activation in the tumour microenvironment. Furthermore, the activation of the JAK/STAT signalling pathway, facilitated by *IFNAR* expression, resulted in the upregulation of *c-Fos* gene, encoding a transcription factor involved in cell proliferation. However, a decrease in signalling via the calcium pathway was observed due to the inhibition of *calcineurin* (*CN*) expression.

Additionally, combination treatment affected the MAPK pathway, leading to the increased expression of *p38*, which could activate apoptosis in cancer cells. Conversely, the increased expression of the transcription factor *AP-1*, involved in the regulation of gene response to various stimuli, including cytokines and growth factors, might stimulate the expression of genes associated with cancer cell proliferation. Moreover, there was a decrease in *PI3K* expression, potentially leading to decreased cell proliferation, survival, and dysregulation of cytoskeletal rearrangement. Interestingly, the upregulation of *p62*, involved in the degradation of misfolded proteins, was observed, potentially due to changes in nutrient status, oxidative stress, and inflammation. Furthermore, the combination treatment increased the expression of *TNFR1*, *NIK*, *IKKα*, and *NFκB*, encoding proteins responsible for activating the transcription of multiple pro-inflammatory cytokines and anti-apoptotic factors, potentially undermining the anticancer effects of the treatment and promoting cancer cell survival.

Furthermore, the combination of cisplatin with high-THC cannabinoid extract #18 significantly increased the expression of genes encoding proteins involved in ER protein processing, as depicted in Figure 11C. This upregulation likely triggered the accumulation of misfolded proteins, ER stress, and unfolded protein response. Notably, there was strong upregulation of proteins involved in dealing with misfolded protein responses, indicating ER stress. However, the expression of genes targeting misfolded proteins for proteasomal degradation was decreased, suggesting further accumulation of misfolded proteins in the ER. Moreover, increased expressions of *PERK* and *IRE1* genes, along with downstream *CHOP* expression, indicated strong signals for apoptosis activation in cancer cells. Additionally, the increased expression of *GADD34*, potentially leading to eIF2α suppression, suggested the inhibition of global translation in treated cancer cells.

Similarly, the combination of cisplatin with extract #18 demonstrated effects similar to extract #18 alone. Notably, there was a significant decrease in the expression of *complexes I* to *V* involved in oxidative phosphorylation (Figure 11D). However, unlike the response to the extract #18 alone, there was no increase in *p53* expression, no decrease in *mPTP*, or any alteration in *cytochrome C* expression. The combination treatment resulted in a more pronounced increase in the expression of *caspases 9* and *3* compared to the extract alone, indicating a stronger activation of apoptosis in the HT-29 cell line.

Furthermore, the combination treatment upregulated certain genes involved in vesicular transport, endo- and exocytosis, potentially facilitating the formation of metastatic niches in the canonical subtype of CRC. Additionally, there was a notable downregulation of PPARγ, which plays a crucial role in insulin sensitization, glucose uptake, and enhancement of energy metabolism in HT-29 cancer cells. An intriguing effect was the inhibition of *NRF* expression, which is involved in cellular respiration and mitochondrial DNA transcription and replication.

As shown in Figure 12A, the combination treatments closely mirrored the effects of high-THC cannabinoid extract #18 alone. Overall, the inhibition of actin cytoskeleton formation suggests a potential decrease in the invasive capacity of cancer cells.

Combining cisplatin with extract #18 resulted in an inhibitory effect on the expression of *TCF*/*LEF*, as illustrated in Figure 12B. These transcription factors play a role in Wnt signalling, which is often dysregulated in the canonical subtype of CRC. Our treatment notably decreased the expression of mRNAs encoding integrins, laminin, and actin, crucial components involved in cell–extracellular matrix interactions. Additionally, there was a decrease in the expression of proteins involved in adherent and desmosomal junction formation, which are vital for cell-to-cell interactions. The disruption of cytoskeletal structures and junction formation may hinder the epithelial-to-mesenchymal transition (EMT) of cancer cells and suppress their invasive potential.

Combining cisplatin with extract #18 enhanced MAPK signalling by increased *HBEGF*, *EGFR*, and *p38* expressions (Figure 12C). While the increased expression of growth factors and their receptors could potentially stimulate tumour cell growth, the substantial upregulation of *p38* coupled with increased *caspase 3* expression might direct cancer cells toward apoptosis. Furthermore, decreased levels of *NFκB* and *IL8* expression suggested a reduction in proinflammatory signalling. Additionally, the decreased expression of *Rac1* and *Cdc-42* likely led to reduced actin reorganization, resulting in decreased cell motility and lowered invasiveness of the HT-29 cell line. However, elevated expression of the gene coding for hepatocyte growth factor receptor cMet might result in stimulation of tumour growth and metastasis. Together with PLCγ, it could promote the cell growth, migration, and proliferation of cancer cells.

In the HT-29 cell line, the combination of cisplatin and extract #18 downregulated *Ras* and *PI3K* expression, major oncogenes often upregulated in multiple cancers (Figure 12D). Moreover, it upregulated the expression of genes encoding p21 and downregulated the expression of transcription factor *cMyc*, potentially leading to cell cycle arrest, the inhibition of proliferation, and the suppression of cell survival mechanisms. However, there was a significant increase in JAK/STAT signalling, which might inhibit apoptosis and promote cell cycle progression through upregulated cyclin D, Bcl-XL, and PIM. Except for the downregulation of *cMyc* gene, these effects closely resembled the action of cisplatin alone.

Figure 12E illustrates the action of cisplatin in combination with extract #18 on multiple pathways involved in the immune system response. The action closely paralleled that of cisplatin alone. There was an increased expression of the genes encoding MHC class II responsible for antigen presentation, ICAM1 enhancing cellular adhesion and pro-inflammatory IL-1β. Moreover, there was an upregulation of *VEGF* expression, responsible for angiogenesis. Additionally, the upregulation of *IL-23* gene might stimulate Th17 cells, inhibiting the cellular immune response of cancer cells. The downregulation of *IL-18* and *IL-8* expressions, responsible for cytotoxic T-cell activation and immune cell chemotaxis, could indicate an attempt by CRC cells to evade immune destruction and activate angiogenesis. These mechanisms were not evident in the action of extract #18 alone.

The combination therapy led to an increase in the expression of various genes involved in the cytokine–cytokine receptor interaction in the CRC cell line HT-29 compared to the DMSO control. As presented in Figure 13A, there was an increase in the mRNA levels of genes encoding chemokines such as CCL22 from the CC and CXC subfamilies, accompanied by a decrease in the expression of the *CXCR4* receptor gene.

On another front, the increased expression of TNF family receptors could potentially activate the extrinsic apoptotic pathway. The diminished expression of TGF-family receptors might hinder the activation of EMT, thereby reducing the invasive potential of the CRC cell line.

The combination of cisplatin with extract #18 applied to HT-29 CRC cell line induced significant changes in the expression of genes involved in activating NK cells against cancer cells (Figure 13B). As previously noted, the upregulation of genes encoding proteins of MHC class I aids in the recognition of cancer cells by the immune system. Additionally, the combinational treatment led to the upregulation of gene encoding ICAM1/2, the cell adhesion molecules, which activate PLCγ, a component of the Ca signalling pathway, facilitating the recognition of cancer cells by NK cells. However, the expression of downstream signalling genes in the Ca signalling pathway, such as *CaN* and *PKC*, was notably decreased. Although *Ras* transcripts were decreased, there was an increase in the expression of *MEK1/2* and *ERK1/2* of genes involved in MAPK signalling, likely subsequently promoting the production of granulocyte-macrophage colony-stimulating factor (GM-CSF).

These effects of the combinatorial treatment might activate immune system surveillance of cancer cells and counteract the effects of cisplatin, particularly in terms of evading immune responses.

## 4. Discussion and Future Perspectives

Cisplatin is commonly used to treat various cancers [64]. It is frequently combined with other chemotherapy drugs [46]. Unfortunately, the use of cisplatin is frequently associated with resistance of cancer to therapy and, thus, cancer progression [44,45]. Cisplatin exhibits a potent pro-apoptotic effect [65,66], which aligned with our findings from the HT-29 CRC cell line.

The mRNA expression analysis conducted on the HT-29 cell line unveiled the multifaceted impact of cisplatin on cancer cells. As illustrated in Figure 14, cisplatin exhibited inhibitory effects on various cellular processes. It altered global translation by targeting aminoacyl-tRNA, rRNA, and spliceosome biosynthetic pathways. Furthermore, cisplatin downregulated several metabolic pathways crucial for cancer cell proliferation, including glycolysis, gluconeogenesis, and oxidative phosphorylation.

A prominent cytotoxic effect of cisplatin was observed through the increased expression of multiple genes involved in the activation and execution of both extrinsic and intrinsic apoptotic pathways. Additionally, cisplatin-induced upregulation of *p21* suggested potential cell cycle arrest at the G1/S phase. However, it is noteworthy that cell cycle arrest may also promote cell survival and is not necessarily advantageous in the context of chemotherapy.

Cisplatin elevated the expression of certain genes involved in the endoplasmic reticulum stress response, which could complement the anticancer actions of cannabinoids. Furthermore, the upregulation of *PDL-1* expression by cisplatin, a recognized target for cancer immunotherapy, presents an opportunity for synergistic effects when combined with immunotherapeutic approaches. This finding highlights a potential milestone in leveraging the synergy between cisplatin and immunotherapy for enhanced anticancer efficacy.

Figure 15 illustrates the potential adverse effects of cisplatin on HT-29 CRC cells as indicated by mRNA sequencing findings. Cytotoxic T-cells play a pivotal role in eliminating cancer cells by engaging death receptors and releasing perforins and granzymes. However, various effects of cisplatin contribute to immune evasion mechanisms. Notably, cisplatin downregulated the expression of *IL18*, crucial for cytotoxic T-cell activation. Conversely, it upregulated *IL4* and *IL10* expression, which can skew the immune response towards humoral immunity and induce regulatory T-cell activation via increased *IL17* and *IL23* expression within the tumour microenvironment.

Interestingly, cisplatin may inadvertently promote cancer progression by enhancing invasion and metastasis through TGF-β signalling and increased exosome formation, facilitating niche preparation in distant organs for successful metastasis [67]. Furthermore, the upregulation of VEGF expression by cisplatin promotes angiogenesis, a hallmark of cancer progression, which coincides with previously published data on cisplatin-resistant oral squamous cell carcinoma cells [68].

In summary, our data showed that cisplatin exhibited contradictory effects on HT-29 CRC cells. While it robustly stimulated apoptosis-related transcripts, the surviving cells may acquire adaptations favouring resistance to subsequent rounds of chemotherapy due to the upregulation of genes associated with tumour progression and evasion of immune surveillance.

Based on our analysis of mRNA expression data, it is evident that high-THC cannabis extract #18 exerts a potent cytotoxic effect on the classical molecular subtype of CRC, as represented by the HT-29 cell line (Figure 16). The tested extract induced cell cycle arrest by downregulating the expression of genes involved in DNA replication and cell cycle progression, while concurrently upregulating tumor suppressor genes such as *p53* and *p21*. Additionally, the inhibition of DNA repair mechanisms, including MMR, BER, and HR pathways, led to genotoxic stress, preventing cancer cells from repairing DNA damage and contributing to cell death. Moreover, the increased expression of *caspase 9*, suppression of oxidative phosphorylation, reduced glucose uptake, inhibition of Ras/PI3K/AKT, and activation of MAPK/p38 pathways further facilitated apoptosis.

The microsatellite unstable subtype of CRC often has mutated genes in the MMR system [69]. In the early stages of tumour development, such genetic changes can promote the formation of driver mutations beneficial for cancer progression. However, in the later stages of carcinogenesis, tumours rely on DNA repair systems to keep them alive due to the huge genotoxic stress encountered by cancer cells. Thus, the inhibition of the MMR system by tested cannabinoid extract can be a double-edged sword in cancer cytotoxicity. It is possible that adding the extract before tumour development might promote carcinogenesis. Still, if added after the CRC becomes a full-blown disease, it might push cancer cells toward death due to the inability to repair irreversible DNA damage. In this case, we tested the cell line representing the classical subtype of CRC. However, it would be interesting to test cannabinoid extracts or cannabinoids alone on microsatellite unstable CRC and see if the treatment is as effective as in the canonical subtype.

Another mechanism by which extract #18 combats cancer is by potentially inhibiting cancer cell invasion and metastasis. Expression analysis revealed decreased the transcription of genes involved in adherent junction formation, dysregulation of the actin cytoskeleton, and downregulation of Rho signalling, which regulates cytoskeletal dynamics, cell migration, and cell cycle progression. The reduced expression of cell adhesion molecules implicated in transendothelial migration suggests the inhibition of cell invasion. Additionally, decreased expression of *TGF-β* gene could impede the EMT, a crucial mechanism in tumour invasiveness. Notably, diminished extracellular vesicle formation, essential for preparing metastatic niches, was also observed. Furthermore, cannabis extract #18 downregulated *COX-2* expression, potentially exerting an anti-inflammatory effect on the CRC cell line.

While our findings strongly support the cytotoxic effect of high-THC cannabinoid extract on HT-29 CRC cells, as corroborated by our MTT results, alterations in certain pathways warrant consideration. For example, there was a pronounced upregulation of protein turnover in the ER, triggering an ER stress response and decreasing expression of genes encoding pro-apoptotic proteins Bax and Bak.

Furthermore, the incorporation of cannabis-derived compounds in clinical research and potential applications entails notable ethical considerations and regulatory challenges. Prioritizing patient safety, securing informed consent, and fostering equitable access are crucial measures. Simultaneously, managing the intricate and evolving legal landscape surrounding cannabis, ensuring product quality standardization, and addressing research gaps are essential aspects of the investigation of cannabinoids and cannabis extracts.

The combination of cisplatin with high-THC cannabis extract #18 elicited alterations in gene expression in the HT-29 cell line. While many changes mirrored those observed with cisplatin or extract #18 alone, some opposing effects of cisplatin on the anticancer action of extract #18 were evident. Furthermore, novel qualitative changes induced by the combination treatment may account for the survival of HT-29 cells under combinational treatment, surpassing the effects of cisplatin or extract #18 individually.

Similarities between the effects of extract #18 alone and in combination with cisplatin included decreased oxidative phosphorylation, reduced translation and transcription, the activation of ER stress, diminished Ras/PI3K and PPARγ signalling, and elevated expressions of p21, suggesting potential cell cycle arrest activation. Additionally, increased levels of caspases 9, 7, and 3 indicated possible apoptosis activation under the combined treatment.

Notably, the combinational treatment countered the pronounced inhibition of cell cycle gene expression and the downregulation of DNA repair pathways observed with extract #18 alone. This reversal might explain the improved survival of the HT-29 cancer cells upon the addition of cisplatin to the extract, as supported by our MTT results. Moreover, there was no inhibition of inflammation; instead, the combination treatment led to the increased expression of genes encoding proinflammatory cytokines, such as IL1β, due to the robust activation of JAK/STAT and NFκB pathways.

Comparing the effects of combining cisplatin with extract #18 revealed persistent unwanted effects of cisplatin alone, including immune response evasion, angiogenesis activation, and microvesicular transport. However, extract #18 potentially mitigated the effects of cisplatin on EMT by inhibiting TGFβ levels. Nevertheless, the combination treatment showed no changes in *PDL-1* expression and no activation of lysosomal hydrolases, contrary to the effects of cisplatin alone.

Surprisingly, the combinational treatment induced more pro-oncogenic changes than the individual treatments, such as the increased expression of genes encoding the cMet oncogene, cyclin D, BclXL, and PIM, indicating the activation of pro-survival pathways. Additionally, increased mRNA expression of proteins activating NK cells could be beneficial, but increased levels of GM-CSF and M-CSF might foster an inflammatory microenvironment around CRC cells.

Recognizing and addressing the antagonistic effects observed in drug interactions is critical for optimizing cancer treatment strategies. It is worth noting that a significant number of cancer patients rely on cannabis compounds to manage pain and mitigate certain adverse effects. To address potential antagonistic effects between cisplatin and high-THC cannabis products, we recommend that healthcare providers advise cancer patients against using cannabis products during platinum-based chemotherapy. This precaution is essential as cannabis use could potentially diminish the effectiveness of chemotherapy, particularly during active treatment. However, additional studies are needed to provide further evidence supporting the idea that this combination treatment demonstrates antagonistic effects in clinical settings.

## 5. Conclusions

In conclusion, the combination of cisplatin and high-THC cannabis extract #18 countered some potential cytotoxic effects on the HT-29 cancer cells observed with cisplatin and extract #18 alone, activating pro-survival mechanisms. Further investigations could focus on purified terpenoids like α-pinene and β-myrcene in combination with THC to elucidate their effects on CRC cell lines. Additionally, while our study provides valuable insights into treatment interactions and underlying molecular mechanisms, confirmatory studies incorporating protein expression, apoptosis assays, and animal models are warranted to validate our findings and guide future research in this field.

## Figures and Tables

**Figure 1 ijms-25-04439-f001:**
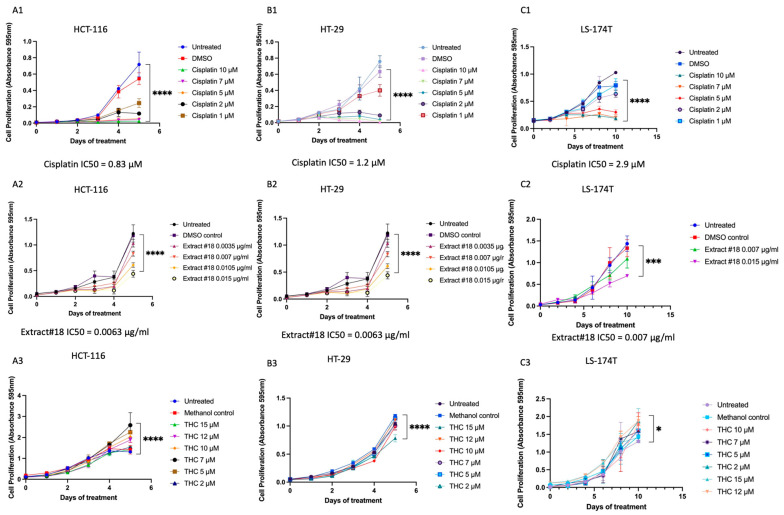
The time and dose-dependent effect of (**A1**–**C1**) cisplatin, (**A2**–**C2**) extract#18, and (**A3**–**C3**) THC on the (**A**) HCT-116, (**B**) HT-29, and (**C**) LS-174T cell lines. Results are shown as means ± SD, conducted in triplicate. Two-way ANOVA was used to calculate time/dose effects. Significant differences between groups are shown as * *p* < 0.05 *** *p* < 0.001, **** *p* < 0.0001. To calculate IC50 concentrations, a nonlinear fit with log(inhibitor) vs. normalized response–variable slope analysis was performed using GraphPad Prism version 9.0.

**Figure 2 ijms-25-04439-f002:**
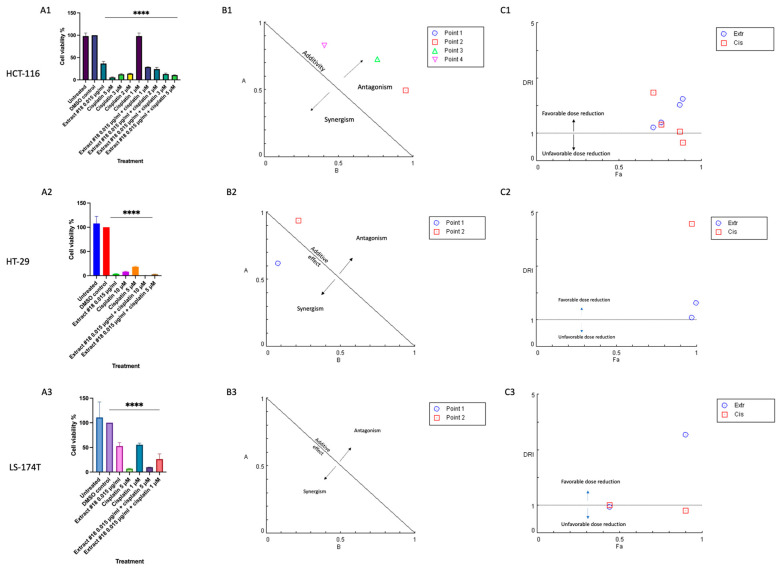
The changes in cell viability in response to extract #18 combined with different doses of cisplatin on the (**A1**) HCT-116, (**A2**) HT-29, and (**A3**) LS-174T CRC cell line based on MTT results. Results are shown as means ± SD, conducted in triplicate. Two-way ANOVA was used to calculate time/dose effects. Significant differences between groups are shown as **** *p* < 0.0001. The effects are shown as normalized isobologram for the combination of extract #18 and cisplatin with normalization of the dose with IC50 to the unity of both x- and y-axes in (**B1**) HCT-116, (**B2**) HT-29, and (**B3**) LS-174T cell lines. Most of the combination points indicated an antagonistic interaction. Normalized isobologram was generated using CompuSyn 1.0 software. Abbreviations: *y*-axis (**A**)—extract #18 (D)_1_/(IC50)_1_; *x*-axis (**B**)—cisplatin (D)_2_/(IC50)_2_; D—dose; Point 1—the combination of extract #18 (0.015 μg/mL) and cisplatin (5 μM); Point 2—the combination of extract #18 (0.015 μg/mL) and cisplatin (10 μM). (**C1**–**C3**) Fa—DRI plot for the combination of extract#18 and cisplatin in different doses in CRC cell lines. Abbreviations: DRI—dose reduction index; Fa—fraction affected by the drug concentration (% of cell growth inhibition/100); Extr—extract #18; Cis—cisplatin. The plot was generated using CompuSyn software.

**Figure 3 ijms-25-04439-f003:**
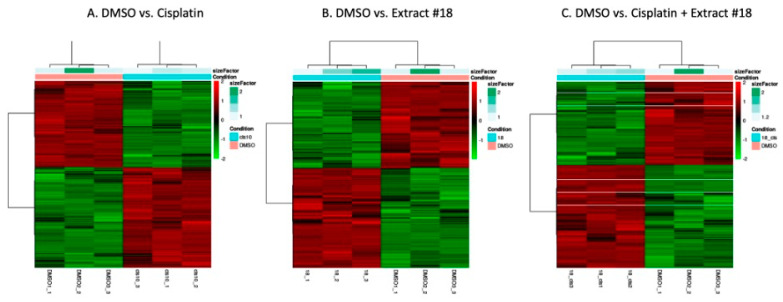
Hierarchical clustering heatmap analysis of the differentially expressed genes with fold change over 1.5 and adjusted *p*–values < 0.05 for (**A**) DMSO versus cisplatin, (**B**) DMSO versus extract #18, and (**C**) DMSO versus cisplatin + extract #18 in the HT-29 CRC cell line. The *x*–axis shows non-supervised clusters between the two treatment groups. The DMSO_1, DMSO_2, and DMSO_3 are independent replicates for DMSO, cis10_1, cis_2, and cis10_3 are replicates for cisplatin 10 μM, 18_1, 18_2, and 18_3 are replicates for extract #18, and 18_cis1, 18_cis2, and 18_cis3 are for combination of cisplatin with the extract #18. The *y*–axis shows differentially expressed genes. The fold changes in upregulated genes are in the red spectrum, and the downregulated genes are represented in the green spectrum. The heatmap was generated using R software.

**Figure 4 ijms-25-04439-f004:**
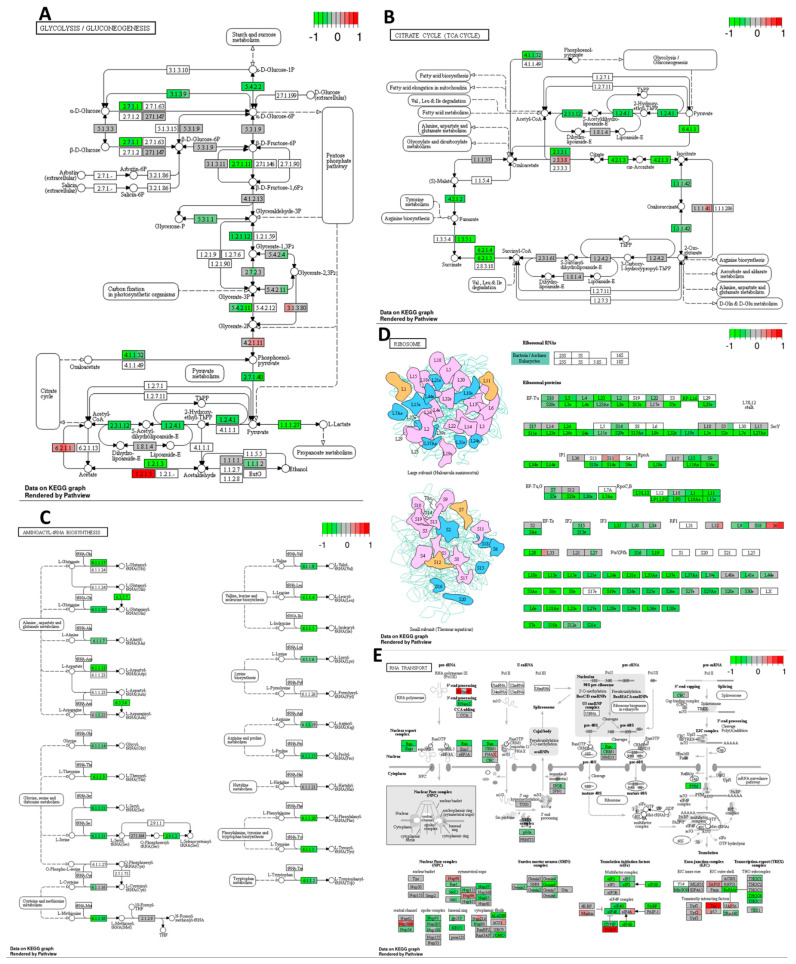
Decreased expression of multiple genes regulating (**A**) glycolysis/gluconeogenesis, (**B**) citrate cycle, (**C**) aminoacyl–T–RNA synthesis, (**D**) ribosome, and (**E**) RNA transport pathways in response to cisplatin 10 μM compared to DMSO control in HT–29 CRC cell line. The genes are coloured according to the difference in expression level between treatment and control for each sample. Red colour shows upregulation and green downregulation relative to the treatment group. Data based on GAGE unidirectional analysis.

**Figure 5 ijms-25-04439-f005:**
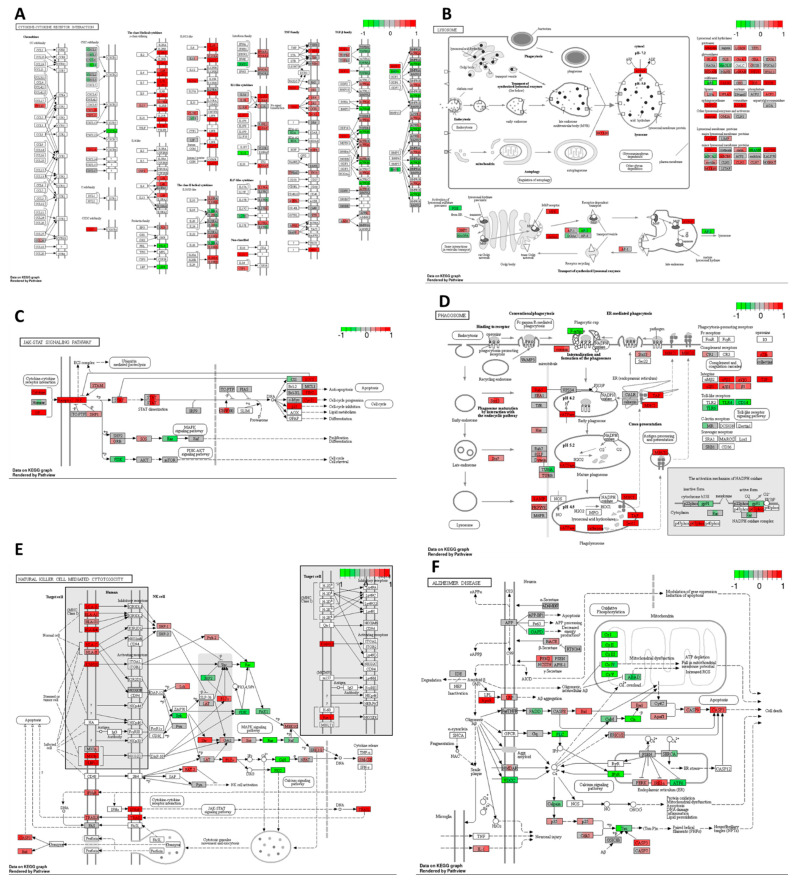
Gene expression changes in GO terms: (**A**) cytokine–cytokine receptor interaction, (**B**) lysosome, (**C**) JAK–STAT signalling pathway, (**D**) phagosome, (**E**) natural killer cell–mediated cytotoxicity, and (**F**) Alzheimer disease pathways in response to cisplatin 10 μM compared to DMSO control in the HT–29 CRC cell line. The genes are coloured according to the difference in expression level between treatment and control for each sample. Red colour shows upregulation and green downregulation relative to the treatment group. Data based on GAGE unidirectional analysis.

**Figure 6 ijms-25-04439-f006:**
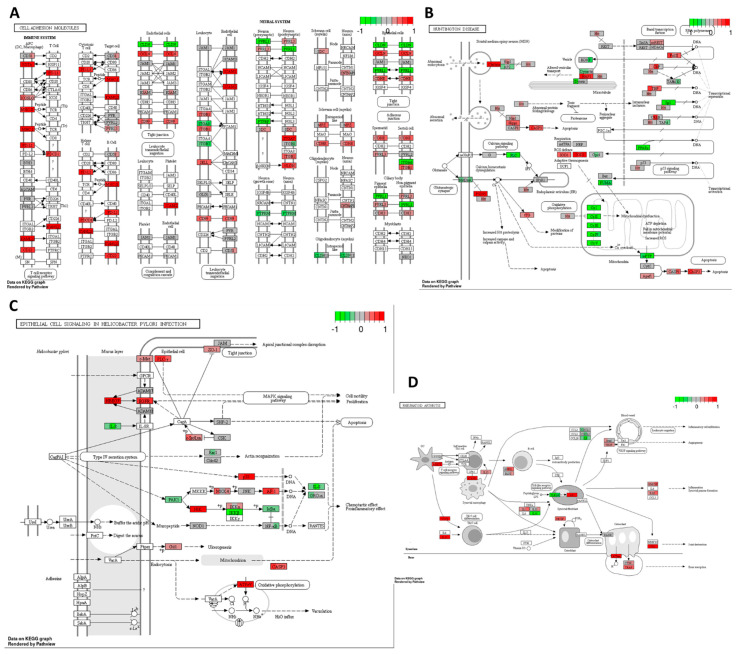
Gene expression changes in GO terms: (**A**) cell adhesion molecules, (**B**) Huntington’s disease, (**C**) epithelial cell signalling in *Helicobacter pylori* infection, and (**D**) rheumatoid arthritis pathways in response to cisplatin 10 μM compared to DMSO control in the HT–29 CRC cell line. The genes are coloured according to the difference in expression level between treatment and control for each sample. Red colour shows upregulation and green downregulation relative to the treatment group. Data based on GAGE unidirectional analysis.

**Figure 7 ijms-25-04439-f007:**
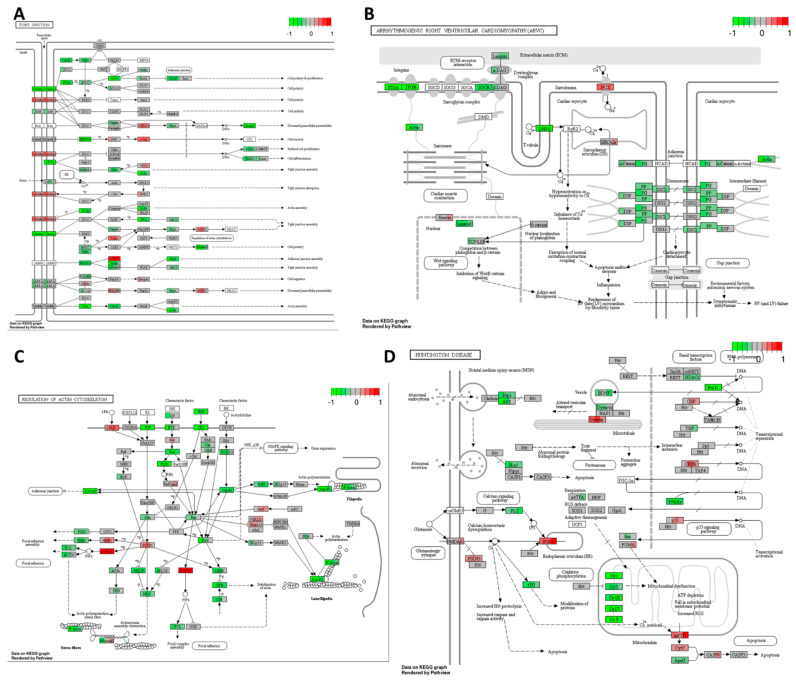
Gene expression changes in GO terms: (**A**) tight junction, (**B**) arrhythmogenic right ventricular cardiomyopathy (ARVC), (**C**) regulation of actin cytoskeleton, and (**D**) Huntington’s disease pathways in response to high–THC extract #18 compared to DMSO control in the HT–29 CRC cell line. The genes are coloured according to the difference in expression level between treatment and control for each sample. Red colour shows upregulation and green downregulation relative to the treatment group. Data based on GAGE unidirectional analysis.

**Figure 8 ijms-25-04439-f008:**
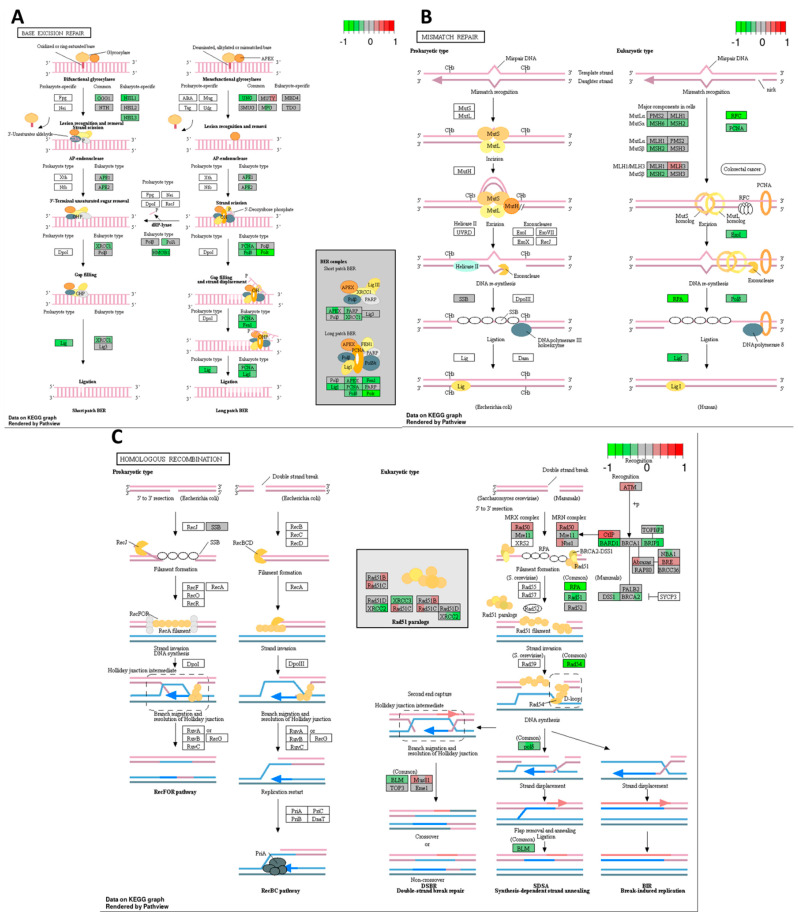
Gene expression changes in GO terms: (**A**) base excision repair, (**B**) mismatch repair, and (**C**) homologous recombination pathways in response to high–THC extract #18 compared to DMSO control in the HT–29 CRC cell line. The genes are coloured according to the difference in expression level between treatment and control for each sample. Red colour shows upregulation and green downregulation relative to the treatment group. Data based on GAGE unidirectional analysis.

**Figure 9 ijms-25-04439-f009:**
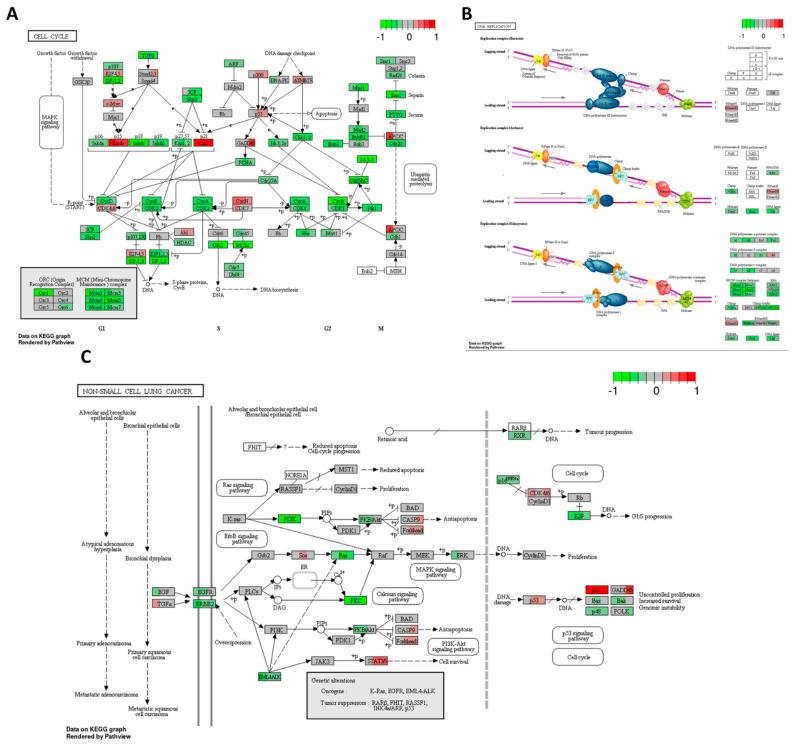
Gene expression changes in GO terms: (**A**) cell cycle, (**B**) DNA replication, and (**C**) non–small cell lung cancer in response to high–THC extract #18 compared to DMSO control in the HT–29 CRC cell line. The genes are coloured according to the difference in expression level between treatment and control for each sample. Red colour shows upregulation and green downregulation relative to the treatment group. Data based on GAGE unidirectional analysis.

**Figure 10 ijms-25-04439-f010:**
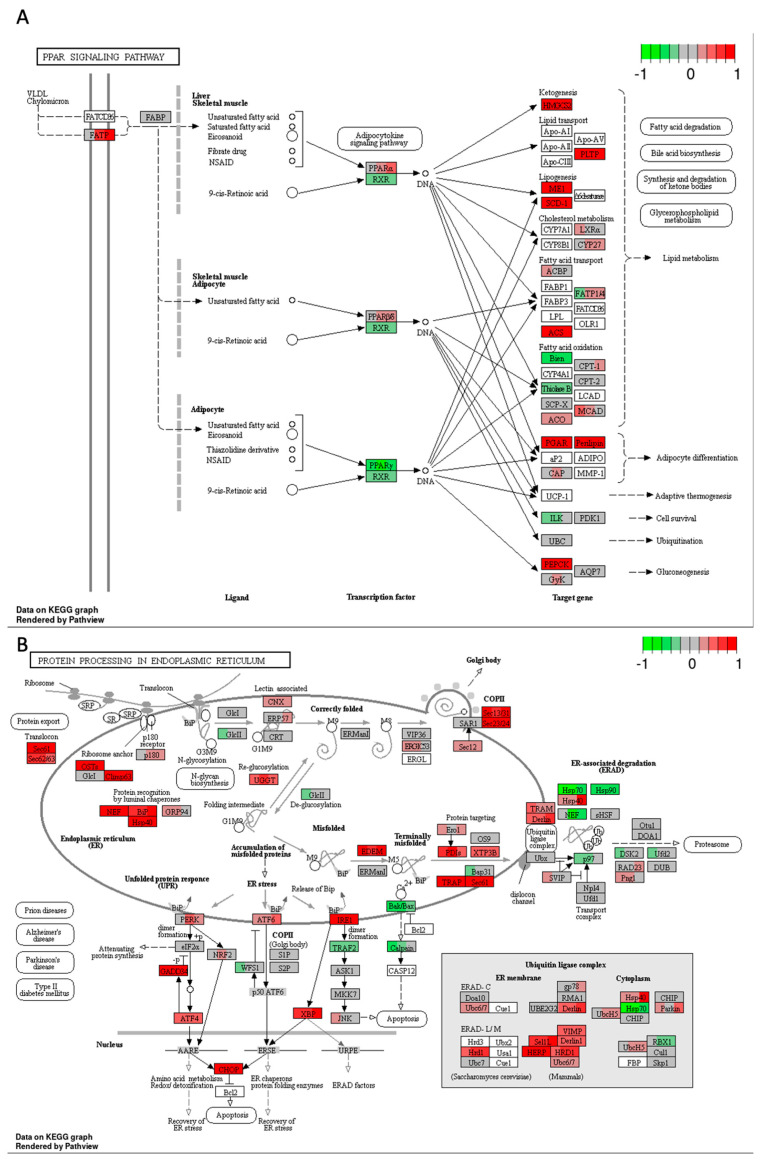
Gene expression changes in GO terms: (**A**) PPAR signalling, and (**B**) protein processing in endoplasmic reticulum in response to high–THC extract #18 compared to DMSO control in the HT–29 CRC cell line. The genes are coloured according to the difference in expression level between treatment and control for each sample. Red colour shows upregulation and green downregulation relative to the treatment group. Data based on GAGE unidirectional analysis.

**Figure 11 ijms-25-04439-f011:**
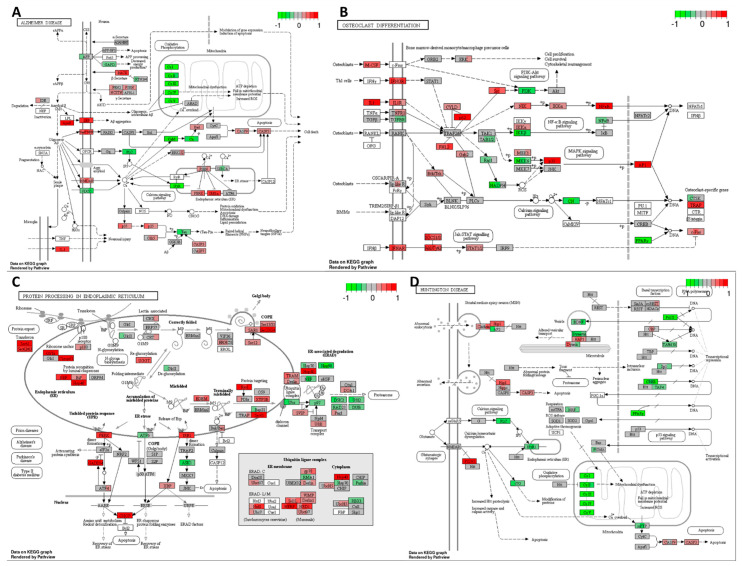
Changed expression of genes regulating GO terms: (**A**) Alzheimer’s disease, (**B**) osteoclast differentiation, (**C**) protein processing in the endoplasmic reticulum, and (**D**) Huntington’s disease in response to high–THC extract #18 and cisplatin combination compared to DMSO in the HT–29 CRC cell line. The genes are coloured according to the difference in expression level between treatment and control for each sample. Red colour shows upregulation and green downregulation relative to the treatment group. Data based on GAGE unidirectional analysis.

**Figure 12 ijms-25-04439-f012:**
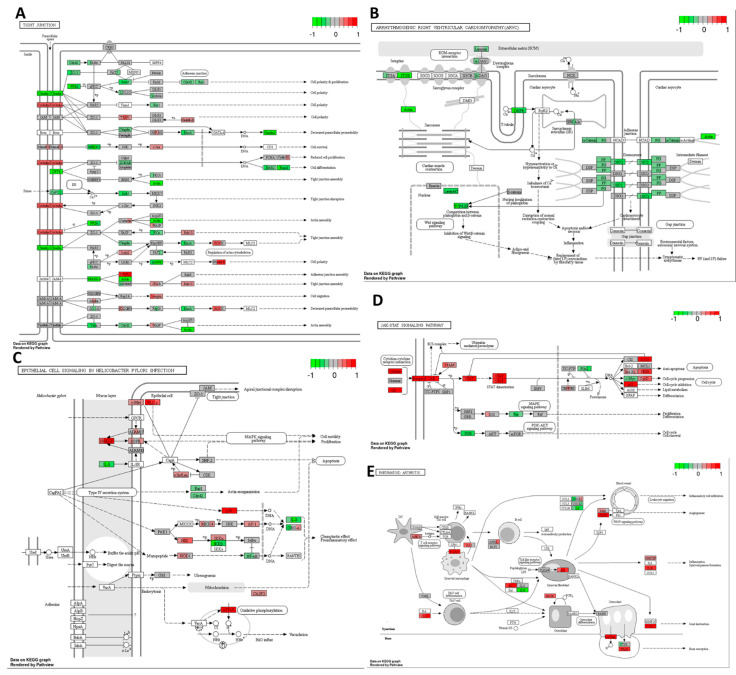
Expression of genes in GO terms: (**A**) tight junction, (**B**) arrhythmogenic right ventricular cardiomyopathy (ARVC), (**C**) epithelial cell signalling in Helicobacter pylori infection, (**D**) JAK–STAT signalling pathway, and (**E**) rheumatoid arthritis in response to high–THC extract #18 and cisplatin combination compared to DMSO in the HT–29 CRC cell line. The genes are coloured according to the difference in expression level between treatment and control for each sample. Red colour shows upregulation and green downregulation relative to the treatment group. Data based on GAGE unidirectional analysis.

**Figure 13 ijms-25-04439-f013:**
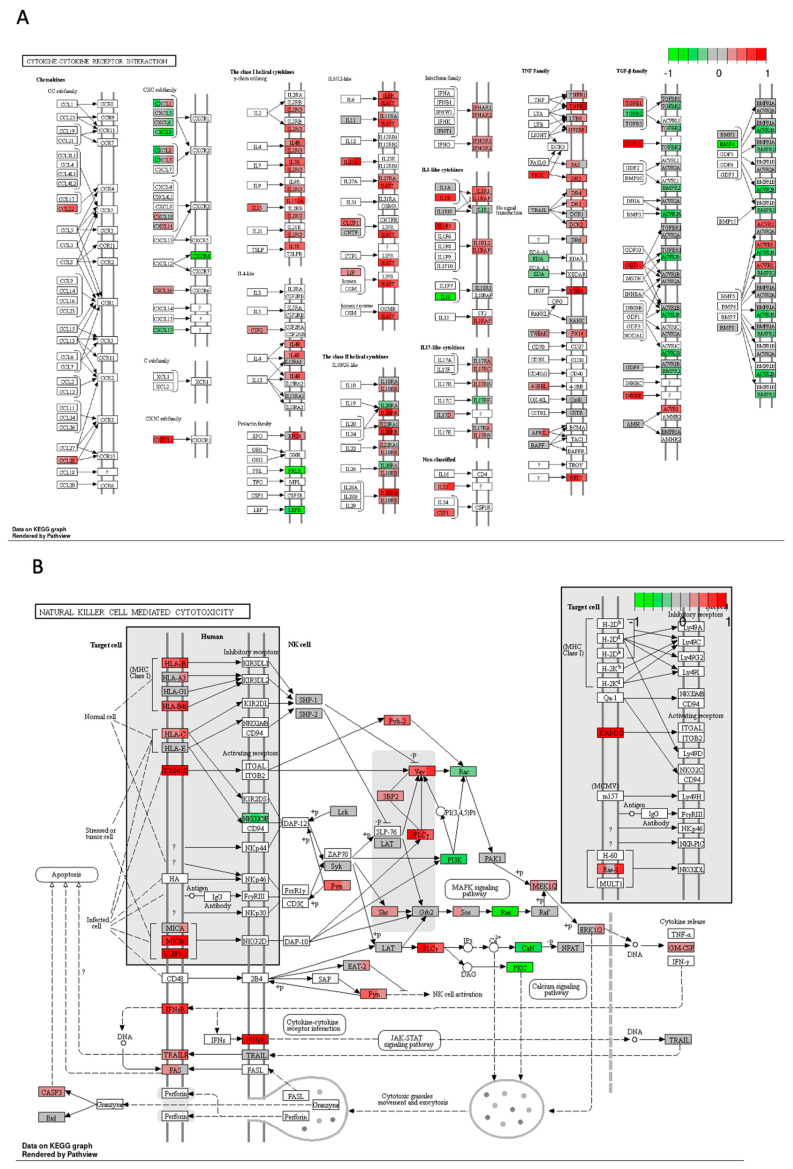
Expression of genes in GO terms: (**A**) cytokine–cytokine receptor interaction, and (**B**) natural killer cell–mediated cytotoxicity in response to high–THC extract #18 and cisplatin combination compared to DMSO in the HT–29 CRC cell line. The genes are coloured according to the difference in expression level between treatment and control for each sample. Red colour shows upregulation and green downregulation relative to the treatment group. Data based on GAGE unidirectional analysis.

**Figure 14 ijms-25-04439-f014:**
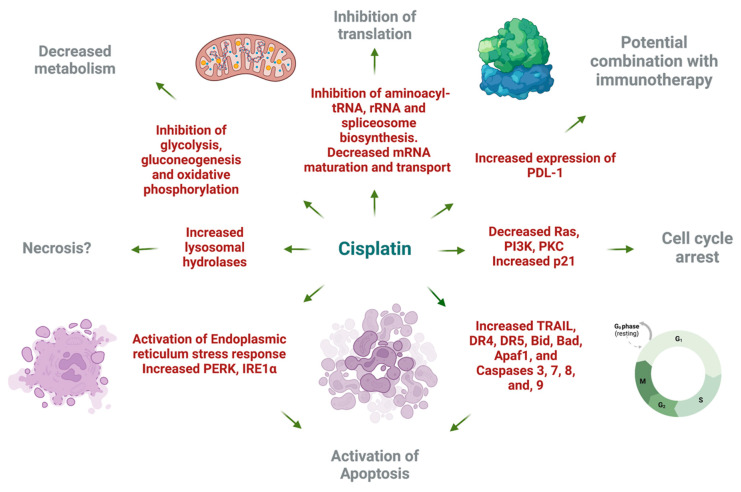
Cytotoxic effect of cisplatin on the HT-29 cell line based on gene expression and pathway analysis. Created with BioRender.com (accessed on 9 April 2024).

**Figure 15 ijms-25-04439-f015:**
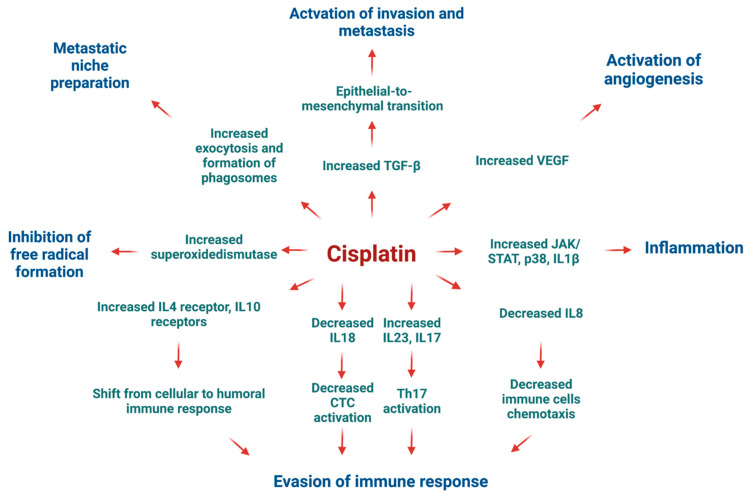
Suggested mechanisms of tumour progression in HT-29 CRC cell line under cisplatin treatment. Created with BioRender.com (accessed on 9 April 2024).

**Figure 16 ijms-25-04439-f016:**
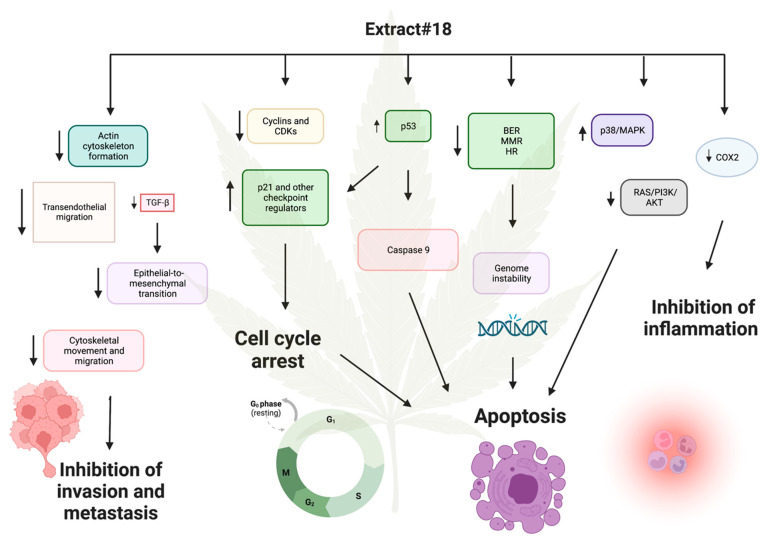
Suggested mechanisms of the cytotoxic effects of high-THC cannabis extract #18 in the HT-29 CRC cell line. Upward –pointing arrows next to the text boxes indicate increased transcription, and arrows pointing downward indicate decreased transcription. Longer arrows between text indicate possible causes and effects. Created with BioRender.com (accessed on 9 April 2024).

**Table 1 ijms-25-04439-t001:** Terpenoid profile of extract #18.

Terpene Levels in Extract #18	Percentage of Dry Weight
alpha-Pinene	0.309
beta-Pinene	0.109
beta-Myrcene	0.42
Limonene	0.035
Terpinolene	0.03
Linalool	0.017
Fenchyl Alcohol	N/A
alpha-Bisabolol	0.003
alpha-Terpineol	N/A
trans-Caryophyllene	0.052
alpha-Humulene	0.035
trans-Nerolidol	0.003
cis-Nerolidol	0
Borneol isomers	N/A
Camphene	N/A
beta-Ocimene	0.003
Fenchone isomers	N/A
Sabinene	N/A
p-Mentha-1,5-diene	N/A
(+)-3-Carene	0.002
alpha-Terpinene	0.07
Eucalyptol	N/A
gamma-Terpinene	0.005
p-Cymene	0.082
Camphor isomers	N/A
Isopulegol	0.002
Caryophyllene oxide	0.107
Valencene	N/A
Geraniol	0.007
Guaiol	0.002
trans-beta-Ocimene	N/A
Sabinene Hydrate	N/A
Isoborneol	N/A
Hexahydrothymol	N/A
gamma-Terpineol	N/A
Geranyl Acetate	N/A
Pulegone	N/A
Nerol	N/A
alpha-Cedrene	N/A
Cedrol	N/A
a-Humulene	0.035
b-Eudesmol	N/A
Total Terpene content	1.328

Abbreviation: N/A—not applicable.

## Data Availability

mRNA sequencing data were uploaded to public repository.

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
