# Peer review of "Targeting Colorectal Cancer: Unravelling the Transcriptomic Impact of Cisplatin and High-THC Cannabis Extract"

_ijms, 2024, doi:10.3390/ijms25084439_

Round 1

Reviewer 1 Report

Comments and Suggestions for Authors

The article “Targeting Colorectal Cancer: Unraveling the Transcriptomic Impact of Cisplatin and High-THC Cannabis Extract” investigated the potential antitumor effect of combinational cisplatin and cannabis extract. While synergistic effects are expected, the authors found that cannabis extract and cisplatin worked antagonistically on various colorectal cancer cell lines. In addition, the authors tried to elucidate the underlying mechanism of the antagonistic effects via RNA-seq. I appreciate the abundant efforts, experiments and analysis made in this work, however, the analysis is too verbose and lack of emphasis. I would recommend minor revision of the manuscript before publication of the manuscript in the present form. Here are my suggestions:

1.      The manuscript is too long and lack of emphasis. I would suggest remarkably shorten the manuscript and put the insignificant information in supplementary materials. Actually even the introduction seems too expatiatory to me.

2.      What is extract #18 from Cannabis sativa? Is it a pure substance or a mixture of unknown substance?

3.      DMSO is used in this work as the control group, please provide the reason of using DMSO instead of PBS as the control group. As far as I know, DMSO has some toxicity on cells, which in my opinion may influence the RNA-seq results.

4.      It is reported in this work that “combinations between high-THC cannabis extract and cisplatin worked antagonistically on different colorectal cancer cell lines.” What is the exact mechanism underlying this antagonistic effects? Too much information is provided in this manuscript that overwhelmed this central and critical point.

5.      Some related literatures should be considered for citation in this manuscript:

1)     Supramolecular Polymerization-induced Nanoassemblies for Self-augmented Cascade Chemotherapy and Chemodynamic Therapy of Tumour. Angew. Chem. Int. Ed. 2021, 60, 17570.

2)     Assessing Gene Expression Related to Cisplatin Resistance in Human Oral Squamous Cell Carcinoma Cell Lines. Pharmaceuticals. 2022, 15(6).

Author Response

The manuscript is too long and lack of emphasis. I would suggest remarkably shorten the manuscript and put the insignificant information in supplementary materials. Actually even the introduction seems too expatiatory to me.

  1.  

We have shortened the manuscript in those parts where it would not affect the quality.

  1. What is extract #18 from Cannabis sativa? Is it a pure substance or a mixture of unknown substance?

It’s a whole-plant cannabis extract from one particular cannabis cultivar. It contains a mixture of cannabinoids terpenes and other substances present in the cannabis plant – the data are presented in the methods.

  1. DMSO is used in this work as the control group, please provide the reason of using DMSO instead of PBS as the control group. As far as I know, DMSO has some toxicity on cells, which in my opinion may influence the RNA-seq results.

DMSO is a standard solvent frequently used for delivering cannabis extracts into cells, and the concentration in the cell culture media did not exceed 0.1%. Moreover, PCA analysis of our RNA seq clustered DMSO with untreated (only media) group, indicating little to no effect of DMSO on gene expression. We also wanted to have the same solvents for both treatments (cisplatin and extract #18).

We did not use PBS, because we were concerned about the possible solubility issues for fatty components of cannabis extracts.

  1. It is reported in this work that “combinations between high-THC cannabis extract and cisplatin worked antagonistically on different colorectal cancer cell lines.” What is the exact mechanism underlying this antagonistic effects? Too much information is provided in this manuscript that overwhelmed this central and critical point.

We added a line in the abstract that we also wanted to show the effects of the cisplatin and extract #18 on the HT-29 CRC cell line separately, not only in combinations. This a critical point of our research.   

  1. Some related literatures should be considered for citation in this manuscript:

1)     Supramolecular Polymerization-induced Nanoassemblies for Self-augmented Cascade Chemotherapy and Chemodynamic Therapy of Tumour. Angew. Chem. Int. Ed. 2021, 60, 17570.

2)     Assessing Gene Expression Related to Cisplatin Resistance in Human Oral Squamous Cell Carcinoma Cell Lines. Pharmaceuticals. 2022, 15(6).

Citations were added to the manuscript.  

Reviewer 2 Report

Comments and Suggestions for Authors

1.      In the abstract, the authors wrote “treatment of various colorectal cancer cell lines”. Instead of writing various colorectal cancer cell lines, authors can write what all the colorectal cancer cell lines are used for this study.

2.      Authors are advised to add quantitative data to the abstract to increase impact.

3.      In the manuscript authors have written “ml and mL”, please follow any one format. It is suggested that the authors follow the same trend throughout the manuscript.

4.      Line 312: authors have written “more than 40 C”. Is it correct?

5.      Figure 2: 1B, 2B, and 3B. The authors have mentioned the figures' y-axis and x-axis as A and B; instead, they are advised to write what they represent.

6. Authors can rewrite the following paragraphs 

Line 14-18, line 48-56, line 83-90, line 137-146, line 191-203, line 213-234, and line 815-821

Comments on the Quality of English Language

Moderate English editing can be done 

Author Response

  1. In the abstract, the authors wrote “treatment of various colorectal cancer cell lines”. Instead of writing various colorectal cancer cell lines, authors can write what all the colorectal cancer cell lines are used for this study.

We made the necessary change.

  1. Authors are advised to add quantitative data to the abstract to increase impact.

We added a few specific genes to the abstract.

  1. In the manuscript authors have written “ml and mL”, please follow any one format. It is suggested that the authors follow the same trend throughout the manuscript.

Changed everything to “ml”.

  1. Line 312: authors have written “more than 40 C”. Is it correct?

Yes, we rewrote the sentence to be less confusing. Changed to “We tested more than forty C. sativa extracts on the HT-29 CRC cell line using cell viability assay…”

  1. Figure 2: 1B, 2B, and 3B. The authors have mentioned the figures' y-axis and x-axis as A and B; instead, they are advised to write what they represent.

Changed

  1. Authors can rewrite the following paragraphs 

Line 14-18 done

line 48-56, done

line 83-90, done

line 137-146, done

line 191-203, done

line 213-234, done

line 815-821 done

Reviewer 3 Report

Comments and Suggestions for Authors

Reviewer comments       

Overall, I found this manuscript somewhat interesting but would have appreciated further investigation into the actual feasibility of using the identified drugs as adjuvant therapy for Cisplatin and High-THC Cannabis Extract.

1.       The manuscript by Viktoriia outlines an approach to address Impact of Cisplatin and High-THC Cannabis Extract during chemotherapy. This is a critical issue, given that many chemotherapeutic agents suppress the immune system. While the use of cisplatin is understandable for cancer therapy, it's not ideal for inhibiting cell death.

2.       The authors present evidence that these drugs inhibit cancer survival pathway. However, essential controls appear to be missing. For instance, it would be crucial to show that apoptotic pathway is unaffected by these drugs, or at a minimum, that other proteins with activity are not inhibited.

3.       I encourage you to clarify whether you are expecting the cisplatin and high-THCto be terpenes and flavonoids.

4.       Please, justify why colon cancer is relevant for demonstrating the effect of flavonoids?

5.       Is it common to present Materials & Methods details only in the Supplemental section.

6.       Cisplatin concentration is given in units of mg/ml while all concentrations for the

tested compounds are given in units of molarity throughout the manuscript. It might be easier for comparison to also give the cisplatin concentration in M

Author Response

Overall, I found this manuscript somewhat interesting but would have appreciated further investigation into the actual feasibility of using the identified drugs as adjuvant therapy for Cisplatin and High-THC Cannabis Extract.

  1. The manuscript by Viktoriia outlines an approach to address Impact of Cisplatin and High-THC Cannabis Extract during chemotherapy. This is a critical issue, given that many chemotherapeutic agents suppress the immune system. While the use of cisplatin is understandable for cancer therapy, it's not ideal for inhibiting cell death.

We explored the action of cisplatin on colorectal cancer cells as platinum salts are used for treating CRC in clinical settings. Since resistance to cisplatin is often an issue in treatment of CRC, we tested cannabis extracts as additives.

  1. The authors present evidence that these drugs inhibit cancer survival pathway. However, essential controls appear to be missing. For instance, it would be crucial to show that apoptotic pathway is unaffected by these drugs, or at a minimum, that other proteins with activity are not inhibited.

Regarding cell death, there was increased expression of caspases, TRAIL, Bid, and Bad under cisplatin treatment. This supports the idea that apoptosis was activated. We did not see similar effects under high-THC cannabis extract treatment. However, in this work, we looked mainly at the RNA expression data, and we addressed the limitations of our work. We mentioned in the discussion section that further experiments need to be done to support our findings.

  1. I encourage you to clarify whether you are expecting the cisplatin and high-THC to be terpenes and flavonoids.

We do not understand this question. We present the terpenoid profile of Cannabinoid extract #18 we used for our study. We do not know, however, whether the difference between THC alone and high-THC extract on their effect on cell viability is due to terpenes, flavonoids or other molecules. This, however, was not the objective of the study. The objective was to find out whether THC alone or THC extract is to be used in combination with cisplatin.

  1. Please, justify why colon cancer is relevant for demonstrating the effect of flavonoids?

We added in the introduction section that some studies that have shown higher potency of whole-plant cannabis extracts over pure cannabinoids on colorectal cancer models. It is known as the entourage effect of cannabis extract that contains cannabinoids, terpenes, flavonoids and other substances present in cannabis extracts. It should be noted, however, that the goal of our work was not to demonstrate the additive effect of terpenes or flavonoids, but rather find out what to be used in the combination with cisplatin.

  1. Is it common to present Materials & Methods details only in the Supplemental section.

The Materials and Method section was placed where recommended by the journal.

  1. Cisplatin concentration is given in units of mg/ml while all concentrations for the

tested compounds are given in units of molarity throughout the manuscript. It might be easier for comparison to also give the cisplatin concentration in M

The concentrations of cisplatin and THC are in mM. Determining the molarity of a plant extract can be challenging because it depends on various factors such as the composition of the extract, the concentration of active compounds, and the molecular weight of those compounds. That is why for the extracts we used a concentration in mg/ml.

Reviewer 4 Report

Comments and Suggestions for Authors

The manuscript titled "Targeting Colorectal Cancer: Unraveling the Transcriptomic Impact of Cisplatin and High-THC Cannabis Extract" reveals a rigorous examination of the interaction between cisplatin, THC, and high-THC cannabis extract in the context of colorectal cancer (CRC) treatment, with a particular focus on the HT-29 CRC cell line. This study contributes significantly to the ongoing exploration of combinatory cancer therapies, offering insights into the molecular mechanisms underpinning the synergistic and antagonistic interactions between these compounds. The article can be accepted with minor revisions:

1. The discussion explores the complexities of drug interactions observed. An expanded discussion on the potential clinical implications of the antagonistic effects observed and strategies to mitigate these effects could be insightful.

2. Given the use of cannabis-derived compounds, a brief discussion on the ethical considerations and regulatory challenges of researching and potentially utilizing these compounds in clinical settings would be pertinent.

Partea superioară a formularului

Author Response

  1. The discussion explores the complexities of drug interactions observed. An expanded discussion on the potential clinical implications of the antagonistic effects observed and strategies to mitigate these effects could be insightful.

Added a small paragraph to the discussion section.

  1. Given the use of cannabis-derived compounds, a brief discussion on the ethical considerations and regulatory challenges of researching and potentially utilizing these compounds in clinical settings would be pertinent.

We added a small section in the discussion regarding these concerns.